# Biological Activities and Chemical Composition of Essential Oil from *Hedyosmum purpurascens* (Todzia)—An Endemic Plant in Ecuador

**DOI:** 10.3390/molecules28052366

**Published:** 2023-03-04

**Authors:** James Calva, Luis Cartuche, Leydy Nathaly Castillo, Vladimir Morocho

**Affiliations:** 1Departamento de Química, Universidad Técnica Particular de Loja (UTPL), Loja 1101608, Ecuador; 2Carrera de Ingeniería Química, Universidad Técnica Particular de Loja (UTPL), Loja 1101608, Ecuador

**Keywords:** *Hedyosmum purpurascens*, germacrene-D, antimicrobial, CMI, anticholinesterase, antioxidant, essential oil

## Abstract

*Hedyosmum purpurascens* is an endemic species found in the Andes of Ecuador and it is characterized by its pleasant smell. In this study, essential oil (EO) from *H. purpurascens* was obtained by the hydro-distillation method with a Clevenger-type apparatus. The identification of the chemical composition was carried out by GC–MS and GC–FID in two capillary columns, DB-5ms and HP-INNOWax. A total of 90 compounds were identified, representing more than 98% of the total chemical composition. Germacrene-D, *ϒ*-terpinene, α-phellandrene, sabinene, *O*-cymene, 1,8-cineole and *α*-pinene accounted for more than 59% of the EO composition. The enantioselective analysis of the EO revealed the occurrence of (+)-α-pinene as a pure enantiomer; in addition, four pairs of enantiomers were found (α-phellandrene, o-cymene, limonene and myrcene). The biological activity against microbiological strains and antioxidants and the anticholinesterase properties were also evaluated and the EO showed a moderate anticholinesterase and antioxidant effect, with an IC_50_ value of 95.62 ± 1.03 µg/mL and a SC_50_ value of 56.38 ± 1.96. A poor antimicrobial effect was observed for all the strains, with MIC values over 1000 µg/mL. Based on our results, the *H. purpurasens* EO presented remarkable antioxidant and AChE activities. Despite these promising results, further research seems essential to validate the safety of this medicinal species as a function of dose and time. Experimental studies on the mechanisms of action are essential to validate its pharmacological properties.

## 1. Introduction

Chloranthaceae is a taxonomic family consisting of herbs, shrubs and aromatic trees of four genera, *Ascarina*, *Chloranthus*, *Hedyosmum* and *Sarcandra* [1]. The species are distributed in the tropical ecosystems of Latin America, Asia and the Pacific [2]. In Ecuador, 16 species of the Chloranthaceae family are recognized [3] and the majority are endemic. The importance of this taxonomic category resides in its primitive origins, applications and distribution patterns, as well as the secretory cells in its leaves and stems, which are common in the Angioespermae family [4]. Most of the chemical constituents of the Chloranthaceae family include sesquiterpenes, terpenoids, coumarins, flavonoids and lignans [5]. In particular, *Hedyosmum* is the most abundant genus of this family, comprising about 40 species located in the mountainous areas of Intertropical America and Southeast Asia [6]. The *Hedyosmum* genus occurs within the geographic limits of cloud forests and in the central Andes, where 50% of its total species are located [7]. Despite their scarce local abundance, these plants have been used in a vast number of applications, such as in ornaments and in the pharmaceutical industry, as fortifiers and antifungal agents, due to the presence of important secondary metabolites, which are the main sources of active ingredients in drugs used to treat respiratory infections and diarrheal diseases [8], In addition, they are used for food purposes, as additives, to improve the organoleptic character of fortifying beverages and liquors [4]. The species of the genus *Hedyosmum* are known to provide biological/pharmacological alternatives, such as antibacterial and antifungal properties, to combat common fungi or infections. In fact, there are studies that demonstrate the veracity of the properties possessed by the essential oils of some species from this genus, such as *H. bonpladianum*, in which the analgesic activity was attributed to isolated glycosidic flavonoids [9].

*Hedyosmum purpurascens* is an herbaceous dioecious plant approximately of 2 to 4 m in height. It possesses brittle branches, purple leaves, a small oval shape with petioles connate around the stem, male inflorescences visible on its last leaves, female inflorescences formed by spikelets located individually in the cavities of the crown, forming small clusters of multiple flowers arranged in the form of cymules and fruits with endocarps and mesocarps of considerable thickness. It grows naturally along the roads of the province of Loja towards the Zamora, Catamayo, Valladolid and Saraguro cantons [2]. There are approximately four endemic species within the Podocarpus National Park, in a small extension of the southern montane evergreen forest of the eastern Cordillera of the Andes [10,11,12].

Despite the age of the species of the *Hedyosmum* genus, only 20 species have been analyzed: *H. traslucidum* [13], *H. brasiliense* [4], *H. angustifolium*, *H. scabrum* [1], *H. arborescens* [14], *H. mexicanum*, *H. bonplandianum* and *H. costaricensis* [15], *H. colombianun* [16], *H. sprucei* [17], *H. glabratum* [18], *H. anisodorum*, *H.uniflorum* [2], *H. luttynii* [2], *H. crenatum*, *H. nutans*, *H. scaberrimum*, *H. maxima*, *H. goudotianum* [19] and *H. racemosun* [20]. Therefore, the aim of this research was to determine the chemical characterization, the enantiomeric distribution and the biological activity of the essential oil of *Hedyosmum purpurascens*. The main purpose was to enrich and promote the knowledge of the vernacular vegetation of our country, including the possible use of this essential oil in the isolation of new compounds and the synthesis of products for food, as well as in medicinal or pharmaceutical applications.

## 2. Results

### 2.1. Physical Properties

The aerial parts of the *Hedyosmun purpurasens* were hydro-distilled to successfully obtain 9.18 mL of a pure essential oil, with a yield of 0.16% (*v/w*). In terms of its sensory characteristics, the EO displayed a yellowish-green color and a slightly sour fresh aroma. The EO revealed a density of 0.87 ± 0.02 g/mL, a refractive index [n20] = 1.4831 ± 0.01 and an average specific optical rotation $\left[\alpha_{514}^{20}\right]$ = −4.78 ± 0.01.

### 2.2. Chemical Constituents of Essential Oil

After the integration of the chromatograms, ninety chemical compounds were identified, representing 97.78%, in the DB-5ms column and 97.18%, in the HP-INNOWax column, of the total constituents (Table 1). The 10 most abundant compounds (representing 64.68 and 59.46% in the DB-5ms and HP-INNOwax, respectively), were Germacrene-D (17.80, 15. 73%), ϒ-terpinene (4.13, 4.53%), α-phellandrene (8.11, 10.84%), sabinene (7.05, 12.33%), ο-cymene (6.62, 3.77%), 1,8-cineole (6.62, 6.62%), α-pinene (5.84, 0.10%), β-pinene (3.29, 0.27%), bisabolene-E-ϒ (2.87, 1.51%) and limonene (2.35, 3.76%).

The EO obtained through the DB-5ms and HP-INNOWAx was characterized by the presence of thirty-two sesquiterpene hydrocarbons (SH), twenty-one monoterpene hydrocarbons (MH), eight oxygenated monoterpenes (OM), seven oxygenated sesquiterpenes (OS) and fifteen other compounds. The SH represented 30.13% and 34.22%, the MH 51.29% and 43.43%, the OM 9.53% and 8.77%, the OS 1.50% and 2.39% and the remaining compounds 5.56% and 10.54%, respectively.

### 2.3. Enantiomeric Composition

When the first enantioselective analysis of the *H. purpurascens* EO was conducted (see Table 2), the (+)-α-Pinene completely occurred as a pure enantiomer and, additionally, four other pairs of optical isomers were found. Among these, the (+)-α Phellandrene presented one of the highest rates of enantiomeric excess, followed by the (+)-o-Cymene, with values of 83.49 ± 0.01 and 79.40 ± 3.41, respectively. Similarly, the lowest enantiomeric excess was presented by the (−)-Myrcene, with a value of 22.67 ± 0.31%.

### 2.4. Antimicrobial Activity

Table 3 shows the minimum inhibitory concentration (MIC) values of the essential oil against the tested microorganisms. The EO of the *H. purpurascens* was found to be inactive at the highest dose tested for the Gram-negative microorganisms and the *Candida albicans* strain. However, for the rest of the strains, the antibacterial capacity proved to be more efficient, particularly for *Staphylococcus aureus* and *Aspergillus niger*, each with a MIC value of 1000 µg/mL. Nevertheless, insufficient relevant outcomes were recorded to propose this EO as an antimicrobial efficient agent for future research.

### 2.5. Antioxidant Activity

The antioxidant potential of the *H. purpurascens* species was evaluated for the DPPH (2,2-diphenyl-1-picrylhydrazyl) and ABTS (2,2-azino-bis(3-ethylbenzothiazoline-6-sulfonic acid) radicals. The analysis was based on the scavenging effect exerted by the EO sample on the 50% radical reduction (SC_50_). Trolox was used as an antioxidant positive control. The essential oil of the *H. purpurasens* essential was not active at the highest dose tested for the DPPH. Instead, it showed a high ABTS uptake capacity, of 56.38 ± 1.96 µg/mL. The results are shown in Table 4.

The Trolox antioxidant capacity was measured according to the same method as that used for the DPPH and ABTS radicals and the SC_50_ was calculated from the dose-response-curve fitting (Figure 1).

### 2.6. Anticholinesterase Activity

Five samples of *H. purpurascens* EO dissolved with MeOH were tested. Donepezil-hydrochloride was used as a positive control, with IC_50_ of 13.6 ± 1.02 µM. The results showed a moderate effect, with an IC_50_ value of 95.62 ± 1.03 µg/mL (Figure 2).

## 3. Discussion

The yield of the *H. purpurascens* EO was estimated based on the mass quantity of the oil obtained and the fresh plant material. The yield obtained is based on the operating conditions under which the whole process was developed, such as the meteorology of plant growth, harvest time, storage of the species, moisture content of the plant, type of hydro-distiller used and steam thermal conditions [69]. On the other hand, the results for the physical properties were in accordance with those of other studies carried out on species of the same genus, such as *H. racemosum*, including the density, with a value of 0.897 g/mL and the refractive index, of 1.4911 [20].

Regarding the qualitative analysis of the EO, more than 95% of the total constituents of the volatile fraction of the sample were identified in the chromatographic columns, DB-5ms and HP-INNOWax. Considering the main compounds found in the oil, it can be inferred that the essential oil of *H. purpurasens* is more closely related to certain studies of species of the same genus; the only difference is in the quantity in the total oil. It was reported that in *H. scabrum*, the major compounds were germacrene-D (13.0%), δ-3-carene (12.1%), α-gurjunene (6.6%), 3′,4′-dimethoxypropiophenone (6.6%), 1,8-cineole (5.7%) and α-phellandrene (2.8%), while in *H. angustifolium* species, the main compounds were α-Pinene (24.0%), β-pinene (23.5%), sabinene (6.4%), linalool (6.1%), 1,8-cineole (3.7%) and germacrene D (3.1%) [70]. In contrast, in a study by Nurliyana et al. [71], the *Chloranthus erectus* EO, a species in the same family as *H. purpurascens*, was analyzed. It was evident that the major compounds in this species are the total opposite of those found in this study, since germacrene-D represented only 0.05%.

The EO composition included SH (30.13%) and MH (51.29%), among which α-pinene, germacrene-D, α-cadinene and α-phellandrene were the most representative. According to Cao et al. [72], these types of compound were considered the main constituents in the essential oils of the Chloranthaceae family, with germacrene-D and cadinene considered some of the main chemotypes. Similarly, the fact that hydrocarbons were found in a greater proportion than the oxygenated hydrocarbons was due to the fact that they are organic compounds with high levels of volatility and, therefore, as they act as regulators of water potential, their extraction occurs more easily. In contrast, in other species, as reported by Stashenko et al. [73], in the EO of *L. alba*, MO predominated (55%); the same was observed for *A. triphylla*, where the MO >70%, since the MWHD (microwave-assisted hydrodistillation) extraction method was used, in which obtaining the EO presents a higher yield [74]. Finally, the presence of other types of compound (5.56%), such as vetivenic acid, E-nuciferol, citronellyl pentanoate, myrtenyl acetate, citronellyl acetate and E-isocroweacin, which have not been reported in other studies, indicated the variation in the chemical compositions of the EO due to the presence of specific chemotypes.

In order to demonstrate the therapeutic properties of *H. purpurascens*, it was found that germacrene-D, the main component of *H. scabrum* and *H. purpurascens*, is a strong antioxidant, given its extracyclic methylene content and its ability to eliminate superoxide anions [75]. Similarly, ϒ-Terpinene possesses biological anti-inflammatory, antimicrobiological and analgesic activities, due to which species containing this compound are used as expectorants and diuretics, as well as in treatment for the relief of muscle pain, fever and asthma [76].

In this study, the antimicrobial activity and Anti-AChE activity of *H. purpurasens* is reported for the first time. Compared with other EOs and with the positive control, our prepared EO had weak antimicrobial activity. Although there are no criteria that are widely used to assess the potency of MIC values, some authors suggested a classification for extracts and essential oils and found that a MIC value between 101 and 500 µg/mL can usually be considered to represent a strong activity [77].

Regarding the cholinesterase inhibitory activity, the *H. purpurasens* EO exerted moderate inhibitory potency, with an IC_50_ value of 56.38 µg/mL; similarly, more recent studies indicated a moderate inhibition of AChE obtained from EOs compared to positive controls, which is in accordance with the reported activity of the EO of *H. brasiliense* of 69.82% at a dose of 1 mg/mL [78]. A study of *H. strigosum* reported a value of 137.6 µg/mL [69]. Many species of the genus *Hedyosmum* have been studied for their antibacterial or antioxidant potential and chemical profile, but little or no inhibitory potential against acetylcholinesterase has been isolated from the EO in this genus. Information on the chemicals used and their inhibitory potential may exist and can be found in the literature.

Although ABTS and DPPH antiradical assays are well recognized and used around the scientific community, the ABTS assay has been confirmed to show better results in detecting the antioxidant properties in foods or in natural matrices, such as extracts or EOs [79]. Therefore, it is noteworthy that our EO displayed better results in ABTS assay; however, further analyses should be carried out to confirm this activity, including the use of another antioxidant assay, such as ORAC or FRAP. The aim of another report will be to analyze the antioxidant properties of the main constituents in essential oils.

## 4. Materials and Methods

### 4.1. Reagents

Dichloromethane, anhydrous sodium sulphate, 2,2′-azinobis-3-ethylbenzothiazoline-6-sulfonic acid (ABTS), 2,2-diphenyl-1-picrylhydrazyl (DPPH), acetylcholinesterase from *Electrophorus electricus*, phosphate buffered saline, Ellman’s reagent (5,5′-dithiobis(2-nitrobenzoic acid), acetylthiocholine iodide and donepezil hydrochloride. The standard aliphatic hydrocarbons were purchased from Chem Service (Sigma-Aldrich, St. Louis, MO, USA), dimethyl sulfoxide was purchased from Merck and helium ultra-pure gas was purchased from INDURA (Guayaquil, Ecuador). All chemicals were of analytical grade and used without further purifications.

### 4.2. Plant Material

Aerial parts of *H. purpurascens* were collected in October 2020 in El Tiro sector, at the border between Loja and Zamora Chinchipe, Ecuador (latitude 3°58′59″ S, longitude 79°08′05″ W). The plant material was harvested under permit MAE-DBN-2016-048 of the Ministry of Environment of Ecuador (MAE) and was authenticated by Dr. Nixon Cumbicus, botanist at Herbarium UTPL. A specimen sample was deposited at the Herbarium of the Universidad Técnica Particular de Loja (HUTPL) with voucher code PPN-as-057.

### 4.3. Isolation of the Essential Oil

Aerial plant material (c.a. 5000 g) was subjected to hydro-distillation immediately after harvesting in a Clevenger-type apparatus for 3 h. The essential oil was dried over anhydrous sodium sulfate to remove the moisture and then stored at −4 °C until use for the biological and chemical assays. The procedure was performed three times.

### 4.4. Physical Properties of Essential Oil

The relative density, refractive index and optical rotation of EO were determined by triplicate at 20 °C. Relative density was determined according to the AFNOR NF T 75-11 method (corresponds to ISO 279:1998), refractive index according to the AFNOR method NF 75-112 (ISO 280: 1998) using the refractometer model ABBE (BOECO, Hamburg, Germany) and, finally, the specific optical determination of rotation using the standard method ISO 592-1998 with an Hannon P-810 automatic polarimeter (Jinan Hanon Instruments Co., Ltd., Jinan, China).

### 4.5. Chemical Characterization of Essential Oil

#### 4.5.1. Sample Preparation

Quantitative and qualitative characterization of EO from *H. purpurascens* required sample preparation of the volatile fractions. The EO was diluted with dichloromethane to reach a concentration of 1%. The samples were used in the chemical analyses described below.

#### 4.5.2. Qualitative and Quantitative Analysis

Qualitative identification was performed using gas chromatography coupled with mass spectrometry (GC–MS): Thermo Fisher Scientific model Trace 1310 gas chromatograph (GC) equipped with Thermo Scientific AI/AS 1300 liquid-sampling automation and model ISQ7000 Mass Spectrometer Single Quadrupole (Waltham, MA, USA). Referring to the experimental conditions of our study, the mass-spectra electronic impact was measured at 70 eV and scanned mass range was set at 40– 400 m/z. Helium was the carrier gas, with a constant flow of 1.00 mL/min. Oven-temperature program was set to an initial temperature of 60 °C for 5 min and then increased to 250 °C at 3 °C/min gradient operation. The ion source temperature was set to 230 °C and the quadrupole temperature to 150 °C. Capillary columns were DB-5 ms (5% phenylmethylpolysiloxane, 30 m × 0.25 mm id, 0.25 μm of film thickness) and HP-INNOWax (polyethylene glycol, film 30 m × 0.25 mm id; thickness 0.2 mm; J & W Scientific, Folsom, CA, USA). The procedure was performed in triplicate.

The aromatic compounds were identified by comparing mass spectra and linear retention index (LRI) with those reported in the literature. The LRI was determined experimentally by Van Den Dool and Krats [80] by injecting a series of homologous C9 to C24 alkanes.

Quantitative analysis of EO was performed using gas chromatography coupled with a flame-ionization detector (GC–FID). The prepared samples were injected under the same analytical conditions as the qualitative GC–MS method with the same column. The percentage of aromatics was determined by comparing the total area of the GC peaks with the identified peaks [39].

#### 4.5.3. Enantioselective Analysis

Enantioselective analysis was performed on the GC–MS instrument mentioned above using a chiral capillary column MEGA-DEX-DET-Beta (diethyl-tert-butylsilyl-β-cyclodextrin; 25 m × 0.25 mm × 0.25 μm) selective analysis. Analytical conditions were the same as for the qualitative oil analysis, except that the oven program was held at 60 °C (5 min) and then increased at a rate of 2 °C to 220 °C for 5 min. The enantiomeric distribution and enantiomeric excess of each enantiomeric pair were determined by comparison with authentic reference compounds.

### 4.6. Anticholinesterase Activiy

The AChE inhibition was measured using the method developed by Ellman et al. [81], with minor modifications, as in Rhee et al. [82]. The AChE inhibition was demonstrated after addition of acetylthiocholine as an enzyme substrate and addition of different concentrations of EO dissolved in methanol. The enzymatic reaction was monitored at 405 nm for 60 min in a microplate reader (EPOCH 2, BioTek, Winooski, VT, USA). Final concentrations of 1000, 500, 100, 50 and 10 µg/mL EO in MeOH were prepared to assess enzyme inhibition. The assay was performed in triplicate in 96-well microplates. Donepezil was used as a positive control. The IC_50_ values were calculated from the progression curves using Graph Pad Prism software.

### 4.7. Antimicrobial Activity

Minimal inhibitory concentration was determined by means of the broth-microdilution method assessing the effect of the EO against seven ATCC reference strains. Three gram-positive bacteria, *Enterococcus faecalis* ATCC 19433, *Enterococcus faecium* ATCC 27270 and *Staphylococcus aureus* ATCC 25923, two gram-negative bacteria, *Escherichia coli* (O157:H7) ATCC 43888 and *Pseudomonas aeruginosa* ATCC 10145 and two fungi, *Candida albicans* ATCC 10231 and *Aspergillus niger* ATCC 6275, were employed. The EO was diluted at concentration of 80 mg/mL and two-fold serial-dilution method was used to achieve concentrations ranging from 4000 to 31.25 μg/mL, with a final cell concentration of 5 × 10^5^ cfu/mL for bacteria, 2.5 × 10^5^ cfu/mL for yeast and 5 × 10^4^ spores/mL for sporulated fungi. The method was developed in a 96-microwell plate and Mueller Hinton II (MH II), for bacteria and Sabouraud broth, for fungi, were used as assay media. The entire method was described in a previous report by our research group [69]. Antimicrobial commercial agents, such as ampicilin 1 mg/mL solution for *S*. *aureus*, *E. faecalis* and *E. faecium*, ciprofloxacin 1 mg/mL solution for *P. aeruginosa* and *E. coli* and, finally, amphotericin B 250 µg/mL for the two tested fungi, were used as positive controls. The DMSO was used as negative control at a maximum concentration of 5%.

### 4.8. Antioxidant Spectrophotometric Analysis

#### 4.8.1. DPPH Assay

The DPPH free-radical-scavenging method was developed by Taipong et al. [83] according to the proposed methodology [84], which was slightly modified to use 2,2-diphenyl-1-picrylhydryl (DPPH-). A working solution in methanol with an adjusted absorbance of 1.1 ± 0.01 (EPOCH 2 microplate reader (BIOTEK, Winooski, VT, USA)) was prepared from 625-micrometer DPPH, methanol-stabilized. The antiradical reactions between different concentrations of EO (two-fold serial dilutions ranging from 1000 to 7.81 µg/mL) and DPPH were carried out as follows: in total, 270 µL of DPPH-adjusted working solution were added in a 96 microplate with 30 µL of EO sample. The reaction was monitored at 515 nm in darkness for 60 min at room temperature. Trolox and methanol were used as positive and blank controls, respectively. Results were expressed as SC_50_ (50% radical scavenging concentration) and calculated from a curve fit of GraphPad Prism v.8.0.1 data. Measurements were performed in triplicate. A calibration standard curve of Trolox was built with concentrations ranging from 1.25 to 70 µM, employing the same method as described above. Linear regression model was obtained by plotting the Trolox concentration vs. absorbance decrease inDPPH radical $\left(y=-0.01408x+1.002;R^{2}=0.9971\right)$.

#### 4.8.2. ABTS Assay

Antioxidant capacity against the ABTS• cation (2,2’-azinobis-3-ethylbenzothiazoline-6-sulfonic acid) was determined as reported by Arnao et al. [85] and Thaipong et al. [83], with slightly modifications, as follows. Briefly, the assay begins with the preparation of a stock solution of free radicals by reacting equal volumes of water solutions of ABTS (7.4 µM) and potassium persulfate (2.6 µM) for 14 h at constant stirring. A working solution was prepared by dissolving an aliquot of the stock solution in methanol until the absorbance reached 1.1 ± 0.02 at 734 nm (EPOCH 2 microplate reader (BIOTEK, Winooski, VT, USA)). The antiradical response was measured by reacting 270 µL of ABTS working solution and 30 µL of EO at the same concentrations employed in DPPH antiradical assay and monitored in darkness at room temperature for 60 min at 734 nm. Trolox and methanol were used as positive and blank controls, respectively. Results are expressed as SC_50_ (50% radical scavenging concentration) and calculated from a curve fit of GraphPad Prism v.8.0.1 data. Measurements were performed in triplicate. A calibration standard curve of Trolox was built with concentrations ranging from 1.25 to 50 µM, employing the same method as described above. Linear regression model was obtained by plotting the Trolox concentration vs. absorbance decreasing of ABTS cation radical $\left(y=-0.02066x+1.152;R^{2}=0.9955\right)$.

## 5. Conclusions

The EO of *Hedyosmum purpurascens* was obtained from the aerial parts of the plant, with a very low extraction yield of 0.16%. The chemical composition of the EO included ca. 90 compounds, of which germacrene-D-, α-phellandrene and sabinene were the most heavily represented. The enantioselective analysis revealed the presence of one optically pure enantiomer and four other pairs of optical isomers, which were (+)-α-pinene, (−) myrcene, (+) α-phellandrene, (+)-limonene and (+) -o-, cymene. The biological activity of the *H. purpurascens* revealed that the EO was inactive for Gram-negative microorganisms. A moderate effect of 95.62 and 56.38 µg/mL was obtained from the AChE enzyme and ABTS assays, respectively. The anticholinesterase and antioxidant effects observed for this species allow us to consider it as a novel candidate for the pharmaceutical industry. This study is a contribution of new knowledge about the native aromatic species of Ecuador.

## Figures and Tables

**Figure 1 molecules-28-02366-f001:**
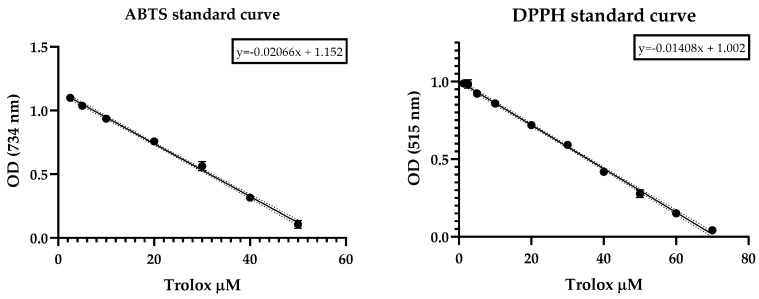
Scavenging effect of Trolox on ABTS and DPPH radicals revealed as a rapid decrease in optical density as a function of the dose.

**Figure 2 molecules-28-02366-f002:**
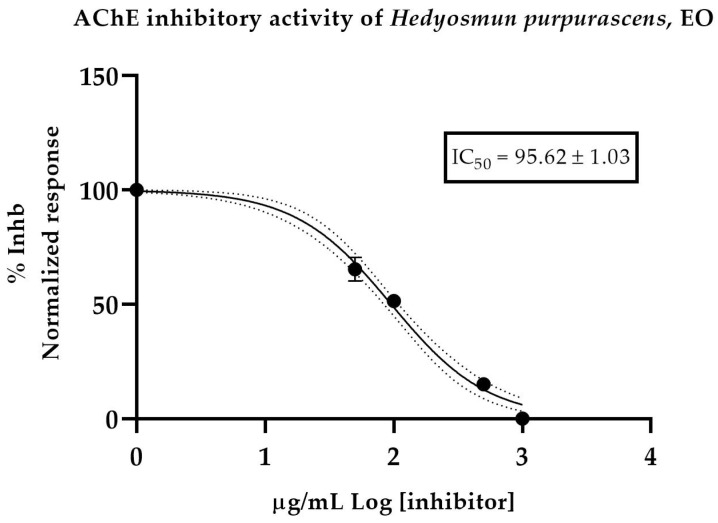
Half-maximum inhibitory concentration of *H. purpurascens* EO against acetylcholinesterase.

**Table 1 molecules-28-02366-t001:** Chemical composition of *H. purpurascens* EO in DB-5ms and HP-INNOWAx columns.

N°	Compuesto	DB-5ms			HP-INNOWAx			
		LRI^a^	LRI^b^	% ± SD	LRI^a^	LRI^b^	Ref	% ± SD
1	α-Thujene	924	924	0.39 ± 0.03	-	-	-	0.01 ± 0.01
2	α-Pinene	932	932	5.84 ± 0.24	1024	1020	[21]	0.10 ± 0.01
3	Camphene	949	945	0.27 ± 0.02	1061	1060	[22]	0.32 ± 0.01
4	Thuja-2,4(10)-diene	953	953	0.81 ± 0.06	1105	1120	[21]	4.53 ± 0.49
5	Sabinene	973	969	7.05 ± 0.36	1119	1123	[23]	12.33 ± 0.32
6	β-Pinene	978	974	3.29 ± 0.21	-	-	-	0.27 ± 0.02
7	Myrcene	990	1003	1.25 ± 0.09	1144	1145	[24]	0.16 ± 0.01
8	Unidentified	993	-	0.15 ± 0.01	-	-	-	0.02 ± 0.01
9	α-Phellandrene	1009	1002	8.11 ± 0.41	1161	1160	[25]	10.84 ± 0.86
10	α-Terpinene	1018	1014	0.75 ± 0.04	1176	1178	[24]	0.81 ± 0.04
11	o-Cymene	1027	1022	6.62 ± 0.27	1185	1187	[26]	3.77 ± 0.01
12	Limonene	1030	1025	2.35 ± 0.16	1195	1193	[27]	3.13 ± 0.39
13	β-Phellandrene	1032	1025	0.43 ± 0.02	-	-	-	0.07 ± 0.01
14	1,8-Cineole	1034	1026	6.62 ± 0.29	1202	1209	[26]	6.74 ± 0.16
15	(Z)-β-Ocimene	1037	1032	0.33 ± 0.02	1203	1245	[28]	0.07 ± 0.11
16	(E)-β-Ocimene	1047	1044	1.05 ± 0.03	1213	1263	[29]	0.17 ± 0.01
17	ϒ-Terpinene	1059	1054	4.13 ± 5.35	1236	1243	[23]	0.46 ± 0.01
18	cis-Sabinene hydrate	1074	1065	0.20 ± 0.03	1252	1450	[30]	1.41 ± 0.23
19	Terpinolene	1087	1086	0.29 ± 0.01	1267	1282	[31]	5.59 ± 1.25
20	α-Campholenal	1096	1122	1.61 ± 0.05	1268	1439	[32]	0.35 ± 0.01
21	Linalool	1105	1095	0.75 ± 0.02	1423	1553	[26]	1.49 ± 0.17
22	Ment-2-en-1-ol -cis-p	1107	1118	0.04 ± 0.01	1465	1571	[32]	0.25 ± 0.01
23	trans-Sabinol	1145	1137	0.04 ± 0.01	-	-	-	0.13 ± 0.01
24	cis-Verbenol	1148	1137	0.56 ± 0.02	1479	1663	[32]	0.91 ± 0.02
25	trans-Verbenol	1152	1140	1.43 ± 0.08	1483	1683	[33]	0.82 ± 0.02
26	Citronellal	1158	1148	0.99 ± 0.05	1479	1488	[34]	0.05 ± 0.01
27	Borneol	1174	1165	0.23 ± 0.02	1500	1721	[35]	0.05 ± 0.01
28	Unidentified	1180	-	0.49 ± 0.05	-	-	-	0.04 ± 0.01
29	cis-Pinocamphone	1183	1172	1.53 ± 0.11	1509	1530	[36]	0.10 ± 0.01
30	Terpinen-4-ol	1186	1174	1.06 ± 0.10	1516	1571	[37]	0.16 ± 0.01
31	ϒ-Terpineol	1203	1199	0.21 ± 0.01	1532	1696	[38]	0.25 ± 0.01
32	Citronellol	1236	1223	0.21 ± 0.02	1536	1597	[39]	2.63 ± 0.05
33	Linalool acetate	1255	1254	0.15 ± 0.01	1546	1548	[37]	0.10 ± 0.01
34	Terpine-4-ol acetate	1297	1299	0.79 ± 0.05	1552	1592	[40]	1.15 ± 0.03
35	Myrtenyl acetate	1315	1324	0.25 ± 0.02	1560	1707	[41]	0.44 ± 0.02
36	α-Cubebene	1346	1348	0.16 ± 0.01	1552	1547	[28]	0.24 ± 0.01
37	Citronellyl acetate	1356	1350	0.09 ± 0.01	1562	1582	[28]	0.13 ± 0.01
38	α-Copaene	1375	1374	0.75 ± 0.05	1568	1535	[22]	0.35 ± 0.01
39	β-Bourbonene	1382	1387	0.13 ± 0.01	1573	1552	[22]	0.54 ± 0.01
40	Unidentified	1385	-	0.03 ± 0.01	-	-	-	0.14 ± 0.02
41	β-Cubebene	1387	1387	0.40 ± 0.03	1594	1573	[39]	0.09 ± 0.01
42	β-Elemene	1389	1389	0.11 ± 0.01	1598	1600	[42]	1.41 ± 0.02
43	E-Caryophyllene	1417	1417	0.76 ± 0.09	1616	1612	[43]	0.75 ± 0.09
44	β-Copaene	1429	1430	0.44 ± 0.03	1625	1613	[44]	0.14 ± 0.01
45	trans-Muurola-3,5-diene	1449	1451	0.15 ± 0.01	1636	1746	[37]	0.22 ± 0.01
46	α-Humulene	1455	1452	0.36 ± 0.03	1655	1662	[23]	0.63 ± 0.01
47	allo-Aromandendrene	1459	1458	0.23 ± 0.02	1662	1661	[45]	0.80 ± 0.01
48	cis-Muurola-4(14),5-diene	1462	1465	0.17 ± 0.01	1676	1630	[46]	2.20 ± 0.03
49	Dauca-5,8-diene	1472	1471	0.17 ± 0.01	1683	1654	[47]	0.59 ± 0.01
50	ϒ-Muurolene	1476	1478	0.57 ± 0.04	1688	1689	[28]	0.83 ± 0.02
51	ϒ-Curcumene	1478	1481	1.02 ± 0.08	1695	1692	[33]	0.21 ± 0.01
52	Germacrene-D	1483	1480	17.80 ± 1.06	1702	1708	[29]	15.73 ± 0.25
53	δ-Selinene	1492	1492	0.52 ± 0.06	1711	1717	[48]	0.22 ± 0.01
54	Bicyclogermacrene	1496	1500	1.46 ± 0.14	1720	1727	[49]	0.33 ± 0.08
55	epi-Cubebol	1498	1493	0.36 ± 0.05	1727	1735	[39]	2.38 ± 0.13
56	α-Muurolene	1501	1500	0.51 ± 0.05	1745	1753	[50]	0.05 ± 0.01
57	α-Bulnesene	1510	1509	0.14 ± 0.04	1754	1505	[51]	2.15 ± 0.05
58	δ-Cadinene	1516	1522	0.40 ± 0.03	1767	1773	[52]	1.19 ± 0.03
59	(E)-ϒ-Bisabolene	1521	1529	2.88 ± 0.25	1772	1769	[38]	1.51 ± 0.03
60	β-Curcumene	1527	1527	1.07 ± 0.12	1787	1734	[53]	0.12 ± 0.04
61	E-Isocroweacin	1534	1553	0.33 ± 0.16	-	-	-	0.13 ± 0.01
62	Unidentified	1536	-	0.13 ± 0.02	-	-	-	0.12 ± 0.01
63	α-Cadinene	1540	1537	0.17 ± 0.01	1848	1807	[54]	0.07 ± 0.01
64	Spathulenol	1584	1577	0.53 ± 0.03	1855	2103	[53]	0.05 ± 0.01
65	Viridiflorol	1592	1592	0.24 ± 0.04	1884	1997	[53]	0.30 ± 0.01
66	Caryophyllene oxide	1594	1582	0.16 ± 0.14	1934	1925	[55]	0.45 ± 0.01
67	Citronellyl pentanoate	1599	1624	0.17 ± 0.10	1973	1880	[56]	0.07 ± 0.02
68	Unidentified	1603	-	0.03 ± 0.01	1996	2077	[57]	0.09 ± 0.02
69	epi-Cubenol-1	1623	1627	0.18 ± 0.01	2004	2347	[30]	0.05 ± 0.01
70	Dill apiole	1633	1620	0.67 ± 0.03	2044	2384	[42]	0.17 ± 0.01
71	epi-α-Cadinol	1637	1638	0.23 ± 0.04	2057	2152	[53]	0.20 ± 0.01
72	allo-Aromandendrene-epoxide	1640	1639	0.27 ± 0.02	2065	2095	[53]	0.06 ± 0.01
73	Vulgarone B	1646	1649	0.63 ± 0.04	2086	2254	[58]	0.43 ± 0.01
74	α-Cadinol	1653	1652	0.43 ± 0.04	2112	2216	[59]	0.16 ± 0.01
75	α-Muurolol	1656	1644	0.34 ± 0.03	2123	2178	[60]	0.34 ± 0.01
76	Intermedeol	1658	1665	0.19 ± 0.02	2142	2264	[61]	0.19 ± 0.02
77	Unidentified	1667	-	0.90 ± 0.11	-	-	-	0.04 ± 0.01
78	Eudesm-7(11)-en-4-ol	1671	1700	0.07 ± 0.01	2193	2320	[33]	0.41 ± 0.01
79	α-Germacra-4(15),5,10(14)-trien-1-ol	1695	1685	0.10 ± 0.01	2197	1686	[62]	0.31 ± 0.01
80	epi-Nootkatol	1699	1699	0.38 ± 0.04	2199	2478	[63]	0.23 ± 0.01
81	Amorpha-4,4-dien-2-ol	1702	1700	0.19 ± 0.15	2204	1690	[62]	0.04 ± 0.01
82	Shyobunol	1704	1688	0.25 ± 0.04	-	-	-	1.07 ± 0.04
83	Zizanal	1710	1697	0.86 ± 0.09	2215	2450	[64]	0.36 ± 0.01
84	Vetiselinenol	1720	1730	0.07 ± 0.04	2215	2245	[64]	0.55 ± 0.02
85	ar-Curcumen-15-al	1726	1712	0.13 ± 0.02	2222	1756	[62]	0.21 ± 0.01
86	E-Nuciferol	1754	1754	0.11 ± 0.01	2225	1758	[65]	0.11 ± 0.01
87	Xanthorrhizol	1762	1751	0.24 ± 0.02	-	-	-	0.42 ± 0.01
88	α-Sinensal	1796	1755	0.16 ± 0.01	2198	1753	[66]	0.34 ± 0.02
89	Vetivenic acid	1810	1811	0.48 ± 0.04	2243	1792	[67]	0.10 ± 0.04
90	Unidentified	2056	-	0.49 ± 0.01	-	-	-	0.36 ± 0.01
	Monoterpene hydrocarbons (%)			51.29	43.43			
	Oxygenated monoterpenes (%)			9.53	8.77			
	Sesquiterpene hydrocarbons (%)			30.13	34.22			
	Oxygenated sesquiterpenes (%)			1.50	2.39			
	Other compounds %)			5.56	10.54			
	Total (%)			98.01	99.35			

LRI^a^, linear retention index calculated; LRI^b^, linear retention index from [68]; Ref, references; %, percentage; SD, standard deviation. Both values were conveyed as means of three determinations.

**Table 2 molecules-28-02366-t002:** Enantiomeric GC analysis of *Hedyosmum purpurascens* EO.

Enantiomeric Compounds	LRI^a^	Enantiomeric Distribution	ee (%) ± SD
(+)-α-Pinene	941	100.00	100 ± 0.01
(+)- Myrcene	993	38.66	22.67 ± 0.31
(−)-Myrcene	994	61.34	
(+)-α-Phellandrene	1029	91.74	83.49 ± 0.01
(−)-α-Phellandrene	1035	8.26	
(+)-Limonene	1053	88.48	76.95 ± 20.46
(−)-Limonene	1058	11.52	
(+)-o-Cymene	1071	89.70	79.40 ± 3.41
(−)-o-Cymene	1075	10.30	

**Table 3 molecules-28-02366-t003:** Antimicrobial activities of *Hedyosmun purpurascens* essential oil.

Microorganism	*H. purpurascens* EO	Positive Control ^a^
	MIC (µg/mL)	
Gram negative rods		
*Escherichia coli*	>4000	1.56
*Pseudomonas aeruginosa*	>4000	0.39
Gram positive cocci		
*Enterococcus faecalis*	4000	0.78
*Enterococcus faecium*	4000	0.39
*Staphylococcus aureus*	1000	0.39
Yeast and sporulated fungi		
*Candida albicans*	>4000	0.098
*Aspergillus niger*	1000	0.098

^a^ Ciprofloxacin for *E. coli* and *P. aeruginosa*; ampicillin for *E. faecalis*, *E. faecium* and *S. aureus*; amphotericin B for *C. albicans* and *A. niger*.

**Table 4 molecules-28-02366-t004:** Half scavenging capacity of *H. purpurascens* EO.

EO	ABTS	DPPH
	SC_50_ (µg/mL—µM *) ± SD	
*H. purpurascens*		
	56.38 ± 1.96	-
Trolox	29.09± 1.05	35.54 ± 1.04

* Half scavenging capacity of Trolox is expressed in micromolar units.

## Data Availability

Not applicable.

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
