# Peer review of "Biological Activities and Chemical Composition of Essential Oil from Hedyosmum purpurascens (Todzia)—An Endemic Plant in Ecuador"

_molecules, 2023, doi:10.3390/molecules28052366_

Round 1
Reviewer 1 Report
The manuscript of James Calva et al. entitled “Biological activities and chemical composition of essential oil... "is an interesting approach complementing the field of study.
Author Response
The manuscript of James Calva et al. entitled “Biological activities and chemical composition of essential oil... " is an interesting approach complementing the field of study.
R: the authors appreciate the reviewer's comments.

Reviewer 2 Report
Dear Authors,
The presented study discusses the chemical composition and biological effects of the prepared essential oil from the Ecuadorian plant Hedyosmum purpurascens. The text is focused on an interesting and current topic.
However, I see fundamental problematic points in the text that need to be modified as part of the revision of the text. The text also needs to be revised in terms of the expression of information and the clarity of the message.
1/ The abstract could be more concise and contain all the main directions of the study and essential results.
2/ Keywords - the list does not reflect the focus of the study (according to the study, I would expect, for example, keywords such as antimicrobial effectiveness, chemical composition, etc.
3/ 2.5.1 it is not clear whether dichloromethane with a concentration of 1% was used for dilution or if the final solution was of this concentration - which should definitely be stated.
4/ Wrongly omitted line between paragraphs in the entire text - see between L126 and L127 etc.
5/ L136 redundant dot
6/ inconsistency in writing titles - font size, see chapter titles, e.g. 2.6 and 2.7
7/ In part 2. MM section, all the methodologies of the performed experiments are missing - for example, there is no methodology for antimicrobial effects, etc. !!
8/ inconsistency in the presentation of information on the labeling of G+ and G- bacteria (!!) - compare the entry on L231 and, for example, in Table 3. I definitely favor the entry given on L231, please edit the entire manuscript.
9/ L231 - "pathogenic" - in all cases it is not a pathogenic bacteria, but also a conditionally pathogenic one.
10/ L232 to L234 I recommend not putting the stock designation in brackets, there is no reason for it.
11/ L235 to L237 the text needs to be rewritten, to better express the information professionally and clearly.
12/ Table 3 - missing statistics for results
13/ All the essentials of the text are not according to the instructions for the authors of Molecules
Author Response
Reviewer 2
The presented study discusses the chemical composition and biological effects of the prepared essential oil from the Ecuadorian plant Hedyosmum purpurascens. The text is focused on an interesting and current topic.
However, I see fundamental problematic points in the text that need to be modified as part of the revision of the text. The text also needs to be revised in terms of the expression of information and the clarity of the message.
1/ The abstract could be more concise and contain all the main directions of the study and essential results.
R: the authors are grateful for the comment, the abstract is improved
2/ Keywords - the list does not reflect the focus of the study (according to the study, I would expect, for example, keywords such as antimicrobial effectiveness, chemical composition, etc.
R: Done
3/ 2.5.1 it is not clear whether dichloromethane with a concentration of 1% was used for dilution or if the final solution was of this concentration - which should definitely be stated.
R: this point was clarified in the text
4/ Wrongly omitted line between paragraphs in the entire text - see between L126 and L127 etc.
R: thanks for the comment, this was corrected
5/ L136 redundant dot
R: Done
6/ inconsistency in writing titles - font size, see chapter titles, e.g. 2.6 and 2.7
R: thanks for the comment, this was corrected
7/ In part 2. MM section, all the methodologies of the performed experiments are missing - for example, there is no methodology for antimicrobial effects, etc. !!
R: thanks, we have added the missing method
8/ inconsistency in the presentation of information on the labeling of G+ and G- bacteria (!!) - compare the entry on L231 and, for example, in Table 3. I definitely favor the entry given on L231, please edit the entire manuscript.
R: thanks for the comment, this was corrected
9/ L231 - "pathogenic" - in all cases it is not a pathogenic bacteria, but also a conditionally pathogenic one.
R: thanks for the comment, this was corrected
10/ L232 to L234 I recommend not putting the stock designation in brackets, there is no reason for it.
R: Done
11/ L235 to L237 the text needs to be rewritten, to better express the information professionally and clearly.
R: thanks for the comment, it was rewritten
12/ Table 3 - missing statistics for results
R: Done
13/ All the essentials of the text are not according to the instructions for the authors of Molecules
R: Done

Reviewer 3 Report
Several comments are shown on the attached manuscript.

Author Response
Reviewer 3
Several comments are shown on the attached manuscript.
R: the authors appreciative of the reviewer's comments, all suggestions have been made.

Reviewer 4 Report
Dear authors,
the subject of your work is certainly unique and the analysis of the volatile organic compounds of such plants and their biological activity is very welcome. It is difficult to follow the thought in some parts of the article, because the sentences are rather confusing. I also have a few comments on the presentation of the results, and I think that by correcting them, your paper will gain more scientific clarity and avoid excessive information that only takes up space but is not very important.
The first figure is, in my opinion, unnecessary, as the structures of already well-known compounds are presented.
Studies on the antioxidant activity of essential oils have not shown very striking results, but perhaps the authors should try other methods of determining this biological activity, such as ORAC, FRAP or others.
Moreover, if such a study is already being carried out, the results should at least be discussed in a few sentences in the Discussion and clearly mentioned in the Conclusions.
Author Response
Reviewer 4
Dear authors,
the subject of your work is certainly unique and the analysis of the volatile organic compounds of such plants and their biological activity is very welcome. It is difficult to follow the thought in some parts of the article, because the sentences are rather confusing. I also have a few comments on the presentation of the results, and I think that by correcting them, your paper will gain more scientific clarity and avoid excessive information that only takes up space but is not very important.
The first figure is, in my opinion, unnecessary, as the structures of already well-known compounds are presented.
Studies on the antioxidant activity of essential oils have not shown very striking results, but perhaps the authors should try other methods of determining this biological activity, such as ORAC, FRAP or others.
Moreover, if such a study is already being carried out, the results should at least be discussed in a few sentences in the Discussion and clearly mentioned in the Conclusions.
R: the authors are grateful for the reviewer's suggestions, which have been considered and incorporated into the article.

Reviewer 5 Report
The manuscript is interesting, original and well written. Although the analytical approach is based on known methodology, the paper provides novel data on the chemical composition and the activities of medicinal plants, in this case by characterizing the essential oil of a rarely studied endemic species from Ecuador. The objective of the study is clear, with the appropriate experimental design to achieve it, and the conclusions are substantiated by the presented results. Still, I would recommend some revisions, which are mostly technical:
Abstract: I cannot agree that the essential oil presented remarkable antioxidant and AChE activities (line 22), as the results suggested either weak or moderate effects. Please, rephrase the sentence.
Introduction: Species names must be italicized (lines 66 and 68).
Materials and methods: Producer details should be given for the polarimeter used (line 103). Reference citation in the text should be according to journal style, by number in brackets – lines 124, 141, 151, 164; as well as in lines 281, 287, 293 (also note that it must be [80], not [820]).
Results: It would be better to give SD values next to the percentage (+/-), and not in separate columns.
Discussion; Conclusions: Compound names in the text should not be capitalized.
Some minor language mistakes need correction – lines 88, 157, 158, 169-170, 221, 274; Table 1 (column #2 heading); Table 3 (table caption).
Author Response
Reviewer 5
The manuscript is interesting, original and well written. Although the analytical approach is based on known methodology, the paper provides novel data on the chemical composition and the activities of medicinal plants, in this case by characterizing the essential oil of a rarely studied endemic species from Ecuador. The objective of the study is clear, with the appropriate experimental design to achieve it, and the conclusions are substantiated by the presented results. Still, I would recommend some revisions, which are mostly technical:
R: the authors appreciate the reviewer's comments.
Abstract: I cannot agree that the essential oil presented remarkable antioxidant and AChE activities (line 22), as the results suggested either weak or moderate effects. Please, rephrase the sentence.
R: done
Introduction: Species names must be italicized (lines 66 and 68).
R: done
Materials and methods: Producer details should be given for the polarimeter used (line 103). Reference citation in the text should be according to journal style, by number in brackets – lines 124, 141, 151, 164; as well as in lines 281, 287, 293 (also note that it must be [80], not [820]).
R: thanks, it is corrected
Results: It would be better to give SD values next to the percentage (+/-), and not in separate columns.
R: done
Discussion; Conclusions: Compound names in the text should not be capitalized.
R: thanks for the comment
Some minor language mistakes need correction – lines 88, 157, 158, 169-170, 221, 274; Table 1 (column #2 heading); Table 3 (table caption).
R: done

Round 2
Reviewer 2 Report
Dear Authors,
I appreciate the improvement of the text based on the previous review process.
However, I still have the following comments on the text (some of the earlier points have not been resolved).
1/ Redundant dot after the title
2/ L14 - missing comma "In this study, ...."
3/ A native English speaker must proofread the text. The text contains many errors and incorrect wording.
4/ It is necessary to reformulate the sentence to L17-18-
5/ It is necessary to reformulate the sentence to L18-21. Please avoid formulating sentences using a colon - in many places throughout the text it is not needed for the sake of expression, and it is also not easy to read.
6/ L24, please reformulate the sentence about the antimicrobial effect (more suitable for a technical report).
7/ L29 small initial letter "anticholinesterase" (what is the reason for the capital letter??) - this appears incorrectly in other places of the text as well - need to go through and correct everywhere.
8/ L123 - wrong gap
9/ chapter 2.7 - needs to be better described to be suitable for the professional text. L152 will only start with the text "Minimal inhibitors....", L153 "...by the broth ...", L153 missing space before the parenthesis, redundant "R" in the designation of all strains used - completely non-standard, there is no reason to use this symbol, L.157 - it is not at all clear which strains are used as model??? (the sentence does not make sense). L158 - 31.25!!!, L158/159 - "...a final cell concentration"!!!, L162 extra dot, L163-164 missing spaces for labeling bacteria, lowercase amphotericin!, L163 avoid colon (different wording !), and ampicillin will be with a small "a".
10/ L162 - "for our research group" - ??? According to the citation, it does not seem to me that it is a publication of the same author's collective or workplace!!! Moreover, the wording of the sentence is inappropriate.
The entire chapter 2.7 must be rewritten according to the conventions of professional texts.
11/ L184, 202 - why is the equation in bold??
12/ L267 - "inactive" - this is not entirely consistent with your results in the table. Some activity was recorded. Of course, it can be presented that it is "negligible," etc.
13/ 3.4 - it is necessary to completely rewrite and supplement the expression of achieved results. The current text does not discuss many results but more about implementation, etc. - inappropriate.
14/ Tab 3 - heading - "essential oil", Gram-negative rods (instead of bacilli - better), missing statistics according to the comment already in the last revision.
15/ L347 - "... our prepared EO has a weak antimicrobial activity"!!
Round 3
Reviewer 2 Report
Dear Authors,
The changes made to the manuscript are satisfactory.